# Preparation and Characterization of Instant Casein Phosphopeptide by Supercritical Fluid Assisted Atomization

**DOI:** 10.3390/foods10071555

**Published:** 2021-07-05

**Authors:** Jian Zhu, Hongsheng Liu, Xingzhe Cai, Wei Wu, Zhiyi Zhu, Long Yu

**Affiliations:** 1Collage of Food Science and Engineering, South China University of Technology, Guangzhou 510640, China; 201910105346@mail.scut.edu.cn (J.Z.); liuhongsheng@scut.edu.cn (H.L.); 202020126228@mail.scut.edu.cn (X.C.); 201721025154@mail.scut.edu.cn (W.W.); fezhuzhiyi@mail.scut.edu.cn (Z.Z.); 2Sino-Singapore International Joint Research Institute, Knowledge City, Guangzhou 510640, China; 3Overseas Expertise Introduction Center for Discipline Innovation of Food Nutrition and Human Health, Guangzhou 510640, China

**Keywords:** casein phosphopeptide, supercritical fluid-assisted atomization, supercritical CO_2_, dissolution rate, bioactivities

## Abstract

Casein phosphopeptide (CPP) has been widely used as micronutrient supplementation for certain populations. However, its solubility performance is far from satisfying. In this work, instant CPP powders with micropore structures were fabricated by supercritical fluid-assisted atomization (SAA) using supercritical CO_2_ fluid (SC-CO_2_) as an atomizing agent. The effects of the processing parameters (temperature, time, and pressure) on SC-CO_2_ absorption rate and dissolution rate were systematically evaluated and studied. The viscosity of the CPP solution increased with increased pressure of SC-CO_2_ as pressure increased its solubility. The processing conditions are optimized as follows: 40 °C, 40 min, and 8.27 MPa, with an SC-CO_2_ absorption rate of about 8 wt.%. The dissolution time of the SAA-CPP powders was significantly decreased from 1800 s to 54 s at room temperature, due to the micropore structures and almost 10 times increase in the specific surface area of SAA-CPP. The bioactivities of the instant SAA-CPP, especially the calcium-binding capacity, were also evaluated and showed no observable difference. Among the four CPPs prepared in different ways in this work, SAA-CPP had better dissolution performance. The results show that SAA technology is a promising way to prepare instant polypeptide powders.

## 1. Introduction

Multi-micronutrient supplementation and food fortification have been shown to be the most cost-effective pathways to alleviate micronutrient deficiencies among pregnant women, infants, and aged people. Casein phosphopeptide (CPP) has been commercialized as micronutrient supplementation in various function and health food formulations, especially in infant formula [1]. CPP can not only promote the absorption and utilization of microelements, such as calcium, iron, and zinc [2,3], but also plays a decisive role in improving immunity [4,5], anticaries [6,7], and antioxidant activities [8]. This is mainly due to its three serine phosphate clusters and two glutamic acid residues [9].

Commercial CPP is the protease hydrolysis product of the milk caseins after being separated, purified, and dried. During its production, the drying method plays a significant role in the dissolution characteristics of the CPP. However, the traditional drying methods such as hot air drying and fluidized bed drying lead to a low dissolution rate: thus, they are detrimental to the application of CPP. Previous studies have shown that vacuum freeze-drying could significantly increase the dissolution rate of products [10]. However, this method is time consuming and energy intensive, so it is not suitable for industrializing the products. Therefore, it is required to develop a new efficient, low-cost, and energy-saving method to prepare instant CPP powders.

Supercritical fluids (SCFs) are termed as substances at a temperature and pressure above their critical points [11], which own unique properties. SCFs possess high density compared with that of a gas state, and high diffusivity, low viscosity, and low surface tension compared with those of liquids. Due to the superior properties of the SCFs, SCF micronization technology has attracted a growing interest in both scientific and industrial communities, and it is widely used in functional material micronization, especially in pharmacy. Conventionally, the technology is classified into four groups: the rapid expansion of supercritical solutions (RESS) [12], the precipitation from gas saturated solution (PGSS) [13], the supercritical antisolvent precipitation (SAS) [14], and the supercritical fluids assisted atomization (SAA) [15,16]. Compared with other SCF technologies, the SAA technology is considered as one of the most promising micronization techniques [17,18].

The SAA technology employs SC-CO_2_ as an atomizing agent that could be used in both organic solvents and water solutions, thus enlarging its applications. Since SC-CO_2_ is non-toxic, non-flammable, low-cost, and owns a low critical point (31.1 °C and 7.38 MPa) [19], the SAA technology has been widely used as a green processing method in the fields of food and medicine. It has been reported that the SAA technology was used to fabricate dried powder from lower viscosity suspensions, such as cyclodextrins microparticles [20], chitosan microparticles [21], and bovine serum albumin microparticles [22]. However, to the best of our current knowledge, no work has been performed on developing the SAA technology to fabricate CPP powders.

In this research, the SAA technology was employed to fabricate instant SAA-CPP using SC-CO_2_ as an atomizing agent. The effect of the SC-CO_2_ processing on the viscosity and porous structure of CPP solution was first studied using a rheometer. The influence of the SAA processing conditions of temperature, time, and pressure on the SC-CO_2_ absorption rate and the dissolution time was further systematically investigated. Then, the morphologies, particle distribution, and the specific surface area of the prepared SAA-CPP were studied. To evaluate the bioactivity of SAA-CPP, we further studied its calcium-binding capacity, total antioxidant capacity, and hydroxyl radical scavenging ability. Meanwhile, the electrical conductivity and foaming properties were also studied. Four kinds of CPP powder, including commercial CPP, spray-dried CPP, vacuum freeze-dried CPP, and SAA-CPP, were compared to study the influence of different drying methods on CPP powders. The physical properties of the four kinds of CPPs prepared in different ways were evaluated, including dissolution time, bulk density, angle of repose, and moisture sorption, the results of which provide valuable information to guide industrialization.

## 2. Materials and Methods

### 2.1. Materials

All the materials used in this study were commercially available: casein phosphopeptide (3.23 wt.% of moisture, 1.83 wt.% of ash, 13.90 wt.% of total nitrogen, purity 99%) was supplied by Guangzhou Lvcui Biological Technology Co. Ltd., (Guangzhou, China); carbon dioxide (CO_2_, purity 99%) was provided by Foshan KODI gas Co. Ltd. (Foshan, China); sodium hydroxide (NaOH) was purchased from Nanjing Chemical Reagent Company (Nanjing, China); sodium dihydrogen phosphate (NaH_2_PO_4_) and calcium chloride (CaCl_2_) were obtained from Shanghai McLean Biochemical Technology Co., Ltd. (Shanghai, China); total antioxidant capacity kit and hydroxyl radical scavenging ability kit were supplied by Nanjing Jiancheng Technology Co., Ltd. (Nanjing, China); all other reagents were of analytical grade and were used without further purification.

### 2.2. Preparation of Instant SAA-CPP

First, the CPP powders were dissolved in distilled water to get a 10 wt.% solution. An integrated SAA apparatus was designed and developed to achieve continuous and highly efficient production. Figure 1 shows the schematic diagram of the SAA apparatus used for the preparation of SAA-CPP powders. It consisted of three feed lines used to transport the SC-CO_2_, CPP solution, and hot air, respectively. CO_2_ was released from the cylinder and pressurized by a high-pressure pump (Guangzhou Gaohe Pump Industry, Guangzhou, China) and then heated in a water bath (Yuhua Instrument Co. Ltd., Foshan, China) to achieve the desired pressure and temperature. The SC-CO_2_ was pumped into a heated reaction kettle to mix with the CPP solution delivered by an injection pump (Guangzhou Gaohe Pump Industry, Guangzhou, China). The mixture was homogeneously mixed at different processing times using a magnetic stirrer. It was then delivered to the drying room and assisted atomization by an injector nozzle with an injection flow rate of 6 mL/min at atmospheric pressure. Hot air at a temperature of about 160 °C was employed to dry the atomized CPP. All the prepared samples were collected in a cyclone separator. Then, the collected SAA-CPP powders were sealed and stored in the dark at 4 °C. All the experiments were carried out in triplicate, and the mean values were used in the data analysis.

The SC-CO_2_ processing conditions of temperature, time, and pressure were investigated, and the corresponding sample codes are listed in Table 1. The commercial CPP without any treatment was tested as a comparison and coded as Untreated CPP. The concentration of CPP solution and the range of the processing conditions were selected based on previous literature [16,23,24] and the critical points of SC-CO_2_. The effect of temperature, time, and pressure on the CPP powders was assessed by CO_2_ absorption rate and dissolution time.

### 2.3. Preparation of Instant SAA-CPP

The spray-dried CPP powders were prepared as follows for comparison. A 10 wt.% solution of commercial CPP (100 mL) without SC-CO_2_ treatment was dried by a spray dryer (EYELA spray dryer SD-100, Tokyo, Japan) with an injection flow rate of 6 mL/min at 160 °C at atmospheric pressure. The vacuum freeze-dried CPP was prepared by the following methods: a 10 wt.% solution of commercial CPP (100 mL) was placed in a vacuum freeze dryer (EYELA, FDU-1110, Tokyo, Japan) overnight to remove the water at −40 °C under high vacuum (ca 0.1 mbar), and the sponge-like CPP was ground to powders before use.

### 2.4. Rheometer

The viscosity of the CPP solution was measured using a TA Instruments Ares-G2 rheometer during the SC-CO_2_ processing in a high-pressure reaction kettle at 2.76, 5.52, and 8.27 MPa, respectively. The measurement was conducted at a temperature ramp from 20 °C to 90 °C with a ramp rate of 5 °C/min and a shear rate of 4 s^−1^.

### 2.5. Determination of SC-CO_2_ Absorption Rate

The SC-CO_2_ absorption rate was measured by weighting the CPP solution before and after SC-CO_2_ processing. Approximately five grams of CPP solution (10 wt.%) was added to the high-pressure reaction kettle. SC-CO_2_ was released from the cylinder and mixed with the CPP solution at different processing conditions (see Table 1). After SC-CO_2_ processing, the pressure of the reaction kettle was carefully reduced within one minute, and the SC-CO_2_-treated CPP solution was immediately weighed and recorded. The SC-CO_2_ absorption rate was calculated as follows:(1)$${\mathrm{SC}\text{-}\mathrm{CO}}_{2}\;\mathrm{absorption}\;\mathrm{rate}\;\left(\%\right)=\frac{m_{2}-m_{1}}{m_{1}}\times 100\%$$
where m_1_ and m_2_ are the weight of untreated CPP solution (g) and SC-CO_2_ treated CPP solution (g), respectively.

### 2.6. Dissolution Rate Measurements

An amount of 1 gram of the prepared CPP powder samples was mixed with 100 mL distilled water in a beaker and stirred using a magnetic stirrer with 500 rpm. The time taken for the CPP powder to completely dissolve was recorded.

### 2.7. Determination of Calcium-Binding Capacity

The calcium-binding capacity of the CPP samples was measured by the pH-state method. NaH_2_PO_4_ (0.096 g) and CPP samples (0.01 g) were dissolved in 100 mL distilled water to obtain the desired concentration of 0.008 mol/L and 0.1 g/L. The mixed solution was heated using the water bath to maintain the temperature at 37 °C. Then, CaCl_2_ (0.088 g) was dissolved in the mixed solution to get a concentration of 0.008 mol/L. The pH of the mixed solution was adjusted to 8.0 in 2 min by a 0.05 mol/L NaOH. This experiment simulated the environment of the terminal of the human small intestine with a pH of 8.0 and temperature of 36–37 °C. Then, the 0.05 mol/L NaOH solution was added to the mixed solution continuously to maintain a pH of 8.0. The consumed volume of the NaOH solution, which could be reflected in the calcium-binding capacity, was recorded every 5 min.

### 2.8. Foaming Properties

The CPP solution (1.00 wt.%, 100 mL) was made to foam using a high-speed homogenizer (Ultra-Turrax T10, IKA, Staufen, Germany) at 10,000~12,000 rpm for 120 s. The foam formation capacity and foam stability of the solution were calculated as follows:(2)$$\mathrm{Foam}\;\mathrm{formation}\;\mathrm{capacity}=\frac{V_{o}}{V_{\mathrm{solution}}}$$
(3)$$\mathrm{Foam}\;\mathrm{stability}=\frac{V_{30\min}}{V_{0}}$$
where V_0_ is the volume of foam at 0 s (mL), V_solution_ is the volume of the initial solution (mL), and V_30 min_ is the volume of foam at 30 min (mL).

### 2.9. Angle of Repose

The angle of repose was determined according to a well-established fixed funnel method [25]. The CPP powders were poured through a funnel onto the graph paper to form a cone. The angle repose was calculated as follows:(4)$$\alpha =\arctan\frac{H}{R}$$
where α is the angle repose (°), H is the height of the CPP powder cone (cm), and R is the radius of the CPP powder circular cone (cm).

### 2.10. Sorption Isotherms

To obtain different relative humidity, a series of over-saturated salt solutions of LiCl (11% RH), CH_3_COOK (23% RH), MgCl_2_ (33% RH), K_2_CO_3_ (43% RH), Mg (NO_3_)_2_ (54% RH), CuCl_2_ (67% RH), NaCl (75% RH), KCl (84% RH), and BaCl_2_ (90% RH) were put in a desiccator at 25 °C overnight to achieve equilibrium. Then, about five grams of CPP powder samples (with an accuracy of 0.0001 g) were spread on a Petri dish and put in the desiccators containing different over-saturated salt solutions at 25 °C. The samples were taken out to measure and record their weight every day till a constant weight was achieved. The moisture absorption rate was calculated as follows:(5)$$\mathrm{Water}\;\mathrm{absorption}\;\left(\%\right)=\frac{\left(M_{1}-M_{0}\right)}{M_{0}}\times 100\%$$
where M_0_ and M_1_ are the weight of the initial sample (g) and the sample after putting it in the desiccator till a constant weight was achieved (g).

### 2.11. Other CPP Characterizations

The surface morphologies of the CPP powders were observed by a scanning electron microscope (ZEISS, Oberkochen, Germany). The particle distributions of the CPP powders were determined by the dry dispersion method using a Mastersizer 3000 (Malvern, UK). The specific surface area of the CPP powders was measured according to the Brunauere-Emmete-Teller (BET) method using a surface area and porosity analyzer (TriStar II 3020, Micromeritics Instrument Corporation, Atlanta, GA, USA). Additionally, the CPP powders were degassed in a vacuum at 70 °C overnight to remove any adsorbates. Total antioxidant capacity and hydroxyl radical scavenging ability were measured by commercially available test kits (Nanjing Jiancheng Technology Co., Ltd., Nanjing, China). The electrical conductivity was measured using a conductivity meter (DDSJ-319L, Shanghai INESA Scientific Instrument Co., Ltd., Shanghai, China) with a CPP solution concentration of 1.00 wt.%. The bulk density of the CPP samples was measured according to GB/T18798.

### 2.12. Statistical Analyses

The results are presented as mean value ± standard deviation. The statistical analyses were performed using SPSS software (IBM, Chicago, IL, USA) for one-way ANOVA with Duncan’s multiple range tests. The significant difference was defined as *p* < 0.05.

## 3. Results and Discussion

### 3.1. Effects of SC-CO_2_ Processing Conditions on CPP Solution

#### 3.1.1. Viscosity

Figure 2 shows the typical viscosity curves of the CPP solutions at different CO_2_ pressures with an increasing temperature detected by a TA rheometer. It is seen that the viscosity of the CPP solutions increased remarkably with increased pressure at all temperatures, indicating that increasing the pressure of carbon dioxide is beneficial to the solubilization of CPP in water. As expected, the viscosities of the CPP solution at different pressures decreased with increased temperature since the intermolecular interaction becomes weak. Therefore, the pressure and temperature of the SC-CO_2_ processing have a significant effect on the CPP solution.

#### 3.1.2. Porous Structure

The CPP suspension rapidly releases gas after high-pressure carbon dioxide treatment in the rheometer, resulting in a rapid expansion of the volumes of bubbles. After SC-CO_2_ treatment, the samples showed as a stable white foam, especially after being treated at high pressure (see Figure 3, upper row). The porous samples were obtained by drying the foams with a vacuum freeze dryer. Figure 3 (bottom row) shows the SEM of the microstructures of the CPP samples before and after SC-CO_2_ treatment in the rheometer. It is seen that the treated samples exhibited a porous structure. Moreover, the CPP foams show denser and smaller pores about 5 µm at higher processing pressure, confirming the enhancement of the SC-CO_2_ absorption by increasing pressure. As expected, the specific surface area of the treated samples is much larger than the original one, which favors the rapid dissolution of the CPP samples. This result theoretically supports the feasibility of using SAA to prepare instant CPP.

### 3.2. Effect of SAA Processing Conditions on CPP

#### 3.2.1. SC-CO_2_ Absorption Rate

Since the SC-CO_2_ plays an essential role in the viscosity and porous structure of CPP solution, the influence of SAA processing conditions on SC-CO_2_ absorption rate of CPP solution was systematically evaluated (see Figure 4a–c). It is seen that the SC-CO_2_ absorption rate was negatively correlated with temperature. The regression equation was y = −0.107x + 12.298, and the correlation coefficient was R^2^ = 0.9992. These results could be interpreted as that with the increase of temperature, the molecules of CO_2_ moved faster and that they could be released more easily from the CPP droplets before spraying. The solubility of gases in liquids is inversely proportional to temperature. A lower temperature could enhance the absorption of SC-CO_2_ of the CPP solution. Therefore, the processing temperature was optimized to 40 °C.

To obtain the SC-CO_2_ absorption rate in the CPP solution, an adequate residence time is required. Figure 4b shows the influence of time (0 min to 60 min) on the SC-CO_2_ absorption rate. The relationship between the absorption rate and the reaction time presented a positive correlation. With the increase of processing time, the SC-CO_2_ absorption rate initially increased sharply and then gradually achieved a constant value in about 40 min. This result indicates that the absorbed SC-CO_2_ in the CPP solution achieved a reversible equilibrium within 40 min. Thus, the processing time was optimized to 40 min for the following experimental work.

The influence of pressure on SC-CO_2_ absorption rate is presented in Figure 4c. When the pressure was less than 6.89 MPa, the SC-CO_2_ absorption rate showed an increasing linear trend with increased processing pressure. An inflection point occurred at 6.89 MPa, and the SC-CO_2_ absorption rate sharply increased from 5.5% under 6.89 MPa to 8.1% under 8.27 MPa. This phenomenon can be explained by the change of CO_2_ state: from a gas state to a supercritical fluid since the critical pressure of CO_2_ is 7.38 MPa. The unique properties of SC-CO_2_ enhanced the interaction of CO_2_ and the CPP molecules. Thus, the SC-CO_2_ absorption rate of the CPP solution increased sharply. With further increased pressure, the absorption rising rate slowed down instead, which was even lower than that under 6.89 MPa. Thus, the processing pressure was optimized to 8.27 MPa for safety concerns and energy-saving strategies.

Generally, decreasing the temperature and increasing the time and pressure can achieve a higher finial SC-CO_2_ absorption ratio under a certain condition. A similar phenomenon was observed for soybean protein [26].

#### 3.2.2. Dissolution Time of SAA-CPP Powders

The effect of temperature, time, and pressure on the dissolution time of the CPP samples was investigated (see Figure 4d–f). It can be seen that the dissolution time of the CPP powders after being processed with SAA decreased significantly from 1800 s to around 60 s, which implies that the dissolution time was shorter by up to 30 times compared with the untreated sample. This result demonstrates that the SAA processing in this work could significantly promote the dissolution rate of CCP powders. Figure 4d shows the influence of processing temperature on the dissolution time. The dissolution time was slightly increased with the increasing temperatures, which showed an opposite tendency with SC-CO_2_ absorption rate. This behavior could be interpreted as follows: With the increasing processing temperature, the surface tension and viscosity of the CPP solution decreased, resulting in a smaller size of the CPP particles. On the other hand, the increasing temperature decreased the absorption of the SC-CO_2_ in the CPP solution, which was proved by Figure 4a. Therefore, the processing temperature of 40 °C was chosen to prepare the instant SAA-CPP powders.

The effect of the processing time on the dissolution time was studied and shown in Figure 4e. In the range of 0 to 40 min, the dissolution time of the CPP powders decreased gradually with the extension of the processing time. However, with a further increase in the processing time, the dissolution time hardly changed, corresponding with the results in Figure 4b. It was mainly due to the SC-CO_2_ absorption in the CPP solution reaching its maximum value after being processed for 40 min.

Figure 4f presents the influence of the processing pressure on the dissolution time of the prepared CPP. It is seen that the dissolution time of the prepared CPP decreased gradually with the increasing processing pressure. Since a higher pressure caused an increased dissolution of SC-CO_2_ in the CPP solutions, accordingly, the primary droplets were more homogeneous. During the depressurization, more CO_2_ was released from the droplet leading to more drastic primary and secondary atomization, which caused a bigger specific surface area of the CPP powder. This result was in agreement with a previous study done by Du et al. [27], which found the specific surface of lysozyme particles increased as the increasing pressure when using SAA processing to produce lysozyme microparticles. A sharp increase in the dissolution rate of CPP powders in the range of 6.89–8.27 MPa was observed as the critical pressure of CO_2_ is 7.38 MPa. This result again indicates the importance of the unique properties of SC-CO_2_ for SAA processing.

Based on the results of CO_2_ absorption and dissolution rate, it can be concluded that the temperature, time, and pressure had a significant influence on the formation of instant CPP powders, and they were optimized to 40 °C, 40 min, and 8.27 MPa. The following experiments of the SAA-CPP powder were conducted based on the optimization parameters.

### 3.3. Characterizations of the SAA-CPP Powder

#### 3.3.1. Morphology Analysis

The morphologies and surface structures of the CPP powders before and after treatment were studied by a scanning electron microscope (see Figure 5). It is seen that the particles of the untreated CPP powders were irregularly blocky-shaped with a reasonably smooth surface and sharp edge (see Figure 5a). The SAA-CPP powders showed a significantly different morphology after being processed with SAA (see Figure 5b). First, the average particle size of the SAA-CPP powders was much smaller than the untreated CPP powders. Secondly, the SAA-CPP powders were spherically shaped with many folds on the surface. There are many hollow balls that can be observed, which may be due to the quick expansion of CO_2_ during spraying processing.

#### 3.3.2. Particle Distribution

The particle size distribution of the untreated CPP powders and the SAA-CPP powders was measured, and the results are presented in Figure 6. The diameter distribution of the untreated CPP powders was in a wide range from 1 μm to 300 μm, with more than 90% of the particles larger than 10 μm. The size of CPP particles was decreased significantly after being processed with SAA. The prepared SAA-CPP powders presented a narrow particle size distribution. Most of them ranged from 1 μm to 20 μm, which is in agreement with the SEM observation (see Figure 5) and can explain the improvement of their solubility performance. Many microparticles with a narrow particle size distribution, such as bovine serum albumin [22] and lysozyme [28], have been prepared by SAA technology. Therefore, the SAA has been considered as an efficient tool for controlling microparticle size.

#### 3.3.3. Specific Surface Area and Pore Structure

The effect of SAA treatment on the surface area is presented in Table 2. It is seen that the specific surface area of SAA-CPP was 2.4 m^2^/g, which is nearly 10 times larger than that of the untreated CPP (0.26 m^2^/g). Meanwhile, the total volume in pores and total area in pores of the SAA-CPP were calculated to be 0.00157 cm^3^/g and 0.963 m^2^/g, respectively, which was largely increased compared with the untreated CPP. These results indicate that the SAA processing could significantly increase the specific surface area and porosity of the SAA-CPP. The results correspond with the results of size distribution measurement and SEM observation. The results can be used to explain the sharp decrease of the dissolution time of the SAA-CPP powders.

#### 3.3.4. Calcium-Binding Capacity

Due to the processing temperature and high pressure employed in this study, maintenance of the bioactivity of the SAA-CPP powder becomes a significant concern. As a commercially available enhancement agent of calcium bioavailability, the calcium-binding capacity of the CPP powders is one of its essential functions. Figure 7 shows the calcium-binding capacity of the untreated CPP and SAA-CPP. As shown, the calcium-binding capacity of the SAA-CPP showed no observed difference with the untreated CPP powders, indicating that the calcium-binding capacity of the SAA-CPP powders obtained in this work could be largely retained.

#### 3.3.5. Physicochemical Properties

To assess the physicochemical properties of the SAA-CPP powders, we further measured their hydroxyl radical scavenging ability, total antioxidant capacity, foam stability, and conductivity (see Table 3). The hydroxyl radical scavenging ability and total antioxidant capacity of SAA-CPP decreased slightly but showed no significant difference with the untreated CPP powders. These results demonstrate that the SAA processing almost had no impact on their hydroxyl radical scavenging ability and antioxidant capacity. This result owes to the relatively mild temperatures employed in SC-CO_2_ processing, the short time of spraying, as well as the absence of organic solvent.

Foamability and foam stability were tested to assess the foaming properties of the SAA-CPP and the results are presented in Table 3. As shown, the foamability of the SAA-CPP was slightly decreased from 60% to 56%. To our surprise, after 30 min the foam stability was significantly decreased from 53.3% to 1.3%. This result was probably due to the removal of the micro emulsifier of the untreated CPP powder after being processed with SAA. The decrease of the foam stability was beneficial to the practical application of CPP powders as micronutrient supplementation, making it more convenient.

The conductivity of the SAA-CPP was also measured to investigate the microstructure of the SAA-CPP. As Table 3 shows, the conductivity value of the SAA-CPP samples scarcely changed from 1043 μS/cm to 1039 μS/cm after being processed with SAA. This demonstrates that the composition of the electrospray ionization of the SAA-CPP solution remains practically unchanged. Therefore, the SAA processing did not disrupt the molecular structure of the SAA-CPP powders, which was identical to the excellent maintenance of bioactivities.

### 3.4. Effect of Different Drying Methods on the CPP

We also investigated the effect of the drying method on the CPP. The four different kinds of CPP powders are evaluated and presented in Figure 8. The dissolution rate is one of the most important physical indicators of instant products, which concerns customers. As shown in Figure 8a, the dissolution time of SAA-CPP was decreased sharply from 1800 s to 54 s, demonstrating that the instant CPP powder was successfully prepared by the SAA apparatus. The dissolution time of spray-dried CPP was 870 s, which was decreased by 2 times than that of the untreated CPP (1800 s). Remarkably, the dissolution time of vacuum freeze-dried CPP was less than that of SAA-CPP powders, which only took 45 s. However, the vacuum freeze-drying consumed a large amount of time and energy, which absolutely hindered its further applications.

In the review of commercial applications, it is also essential to consider the physical properties of the CPP powders. Thus, we further compared the bulk density, angle repose, and moisture content of the four CPP powders. Bulk density estimates the biomass per unit area and projects transportation and logistics, as well as cost reduction [29]. Figure 8a shows that the bulk density of the untreated CPP powders was 0.72 g/mL. In contrast, the bulk density of the CPP powders was decreased significantly after drying in different ways, which was unfavorable to the packaging and transportation. The bulk density of the spray-dried CPP was 0.17 g/mL, which was the lowest of the four types of CPP. The bulk density of the SAA-CPP powders was 0.3 g/mL, which was slightly lower than that of vacuum freeze-dried CPP (0.34 g/mL). However, the bulk density of the vacuum freeze-dried CPP powders was 0.14 g/mL before grounding due to its spongy structure.

The flowability of the CPP powders was assessed by the angle of repose, presented in Figure 8b, which is the angle between the CPP powder surface and the horizontal surface. An angle of repose smaller than 30° describes excellent powder flowability, 31–35° designates good flowability, 36–40° indicates fair flowability, and θ > 40° means poor flowability [30]. As shown, the angle of repose of the untreated CPP powders was 27.2°, which specified that it possessed excellent flow property. However, the angle of repose of the spray-dried CPP was 37.3°, which describes fair flowability. Additionally, the vacuum freeze-dried CPP powders and SAA-CPP powders present good flow property since their angle of repose were 31.3° and 32.9°, respectively.

The inter-particulate interaction, including friction and adhesion, was considered as an essential factor affecting the bulk density and flowability of the powders [31]. The results show the bulk density and flowability of the treated CPP powders were both decreased, demonstrating the decrease of the inter-particulate interaction of the CPP powders. These behaviors were attributed to the increase of the surface roughness of the CPP powders after drying in different ways, which was consistent with the result of Figure 7. However, the decrease of the bulk density and flowability was not that substantial and was acceptable in commercial applications.

The moisture sorption isotherms were measured to study the interaction between water and the CPP powders and can also be used to assess the storage stability of the CPP powders. Figure 8c presents the adsorption plots of water vapor of the four different samples at room temperature. As shown in Figure 8c, the moisture absorption isotherm of the four CPP powders showed the same tendency, in that the moisture absorption increased with the increasing relative humidity due to their hydrophilicity. A dramatic increase in the moisture absorption was observed at high relative humidity (aw > 0.6 at 25 °C). Similar results have been reported in previous studies for many food powders, such as pure orange juice powder [32], tamarind pulp powder [33], and tomato pulp powder [34]. These behaviors can be attributed to the hydrophilic nature of carbohydrates and protein contained in the powders. The moisture sorption isotherms indicate that the moisture content of the four CPP powders at a constant water activity was as follows: untreated CPP < spray-dried CPP < SAA-CPP < vacuum freeze-dried CPP. These results could be due to the structure of their particles and demonstrates their ability to capture the moisture, which is corresponds with their dissolution time (see Figure 8a).

All in all, when compared with the untreated CPP, the solubility performance of the three kinds of drying CPP was improved significantly with a slight decrease of physical properties. Both the SAA-CPP and vacuum freeze-dried CPP showed a remarkable dissolution rate within one minute. However, the vacuum freezing-drying method is time consuming and energy intensive which leads to a high production cost. The SAA method studied in this work can achieve continuous and highly efficient production with low cost, which is a promising way to process CPP powders.

## 4. Conclusions

This work used an integrated SAA apparatus to fabricate instant SAA-CPP powders. First, the viscosity of the CPP solution was studied using a rheometer, confirming that the SC-CO_2_ processing pressure has a great effect on the solubility of CPP and the porous structures of CPP foams. The processing conditions of SAA technology were further systematically investigated and optimized to 40 °C, 40 min, and 8.27 MPa, where a CO_2_ absorption of 8% and a dissolution time of 54 s were obtained. Compared with the untreated CPP, the particle size of the SAA-CPP decreased significantly and presented a narrow distribution from 1 μm to 20 μm. Meanwhile, the specific surface area of the SAA-CPP increased nearly 10 times, explaining the dramatic decrease of dissolution time. The bulk density and flowability of the SAA-CPP decreased slightly even with smaller particles since it contains porous structures. Furthermore, SAA-CPP increased the ability to capture the moisture. The physicochemical properties of SAA-CPP, including the calcium-binding capacity, hydroxyl radical scavenging ability, total antioxidant capacity, foamability, and conductivity, showed no significant difference, revealing the retention of its bioactivities and chemical structure. There was a significant decrease in its foam stability due to the removal of the micro emulsifier, which is beneficial to the application of SAA-CPP.

## Figures and Tables

**Figure 1 foods-10-01555-f001:**
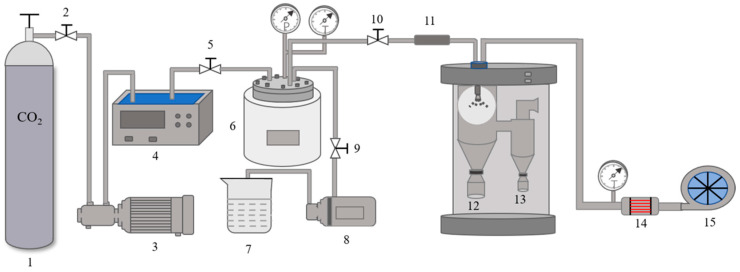
The schematic diagram of the SAA apparatus used for the preparation of instant CPP powder (1, CO_2_ cylinder; 2, 5, 9, 10, valve; 3, high-pressure pump; 4, water bath; 6, high-pressure reaction kettle; 7, sample solution; 8, injection pump; 11, flow restrictor; 12, drying room; 13, cyclone separator; 14, temperature control display; 15, hot air).

**Figure 2 foods-10-01555-f002:**
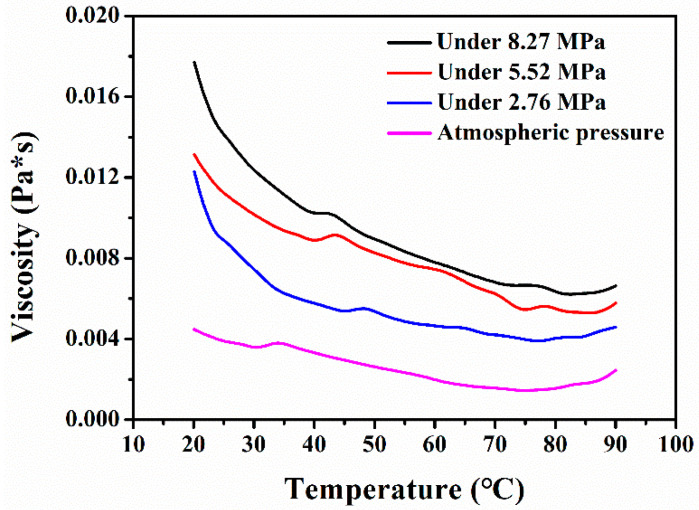
The viscosity of the CPP solutions at different SC-CO_2_ processing pressures with increased temperature.

**Figure 3 foods-10-01555-f003:**
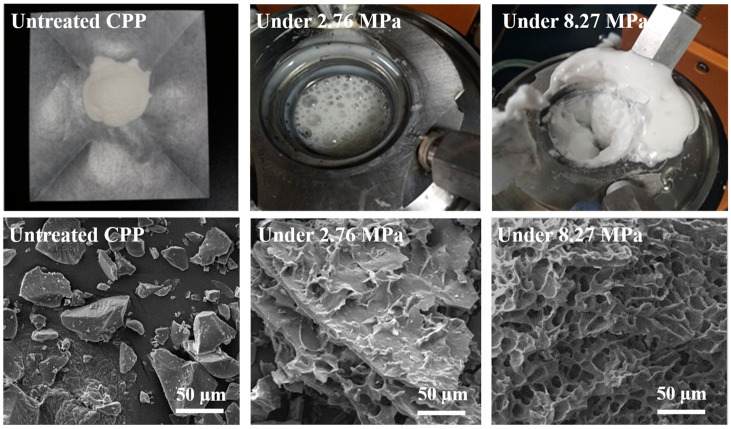
The photographs (**upper row**) and SEM microstructures (**bottom row**) of the CPP: unprocessed, processed under 2.76 MPa, and processed under 8.27 MPa.

**Figure 4 foods-10-01555-f004:**
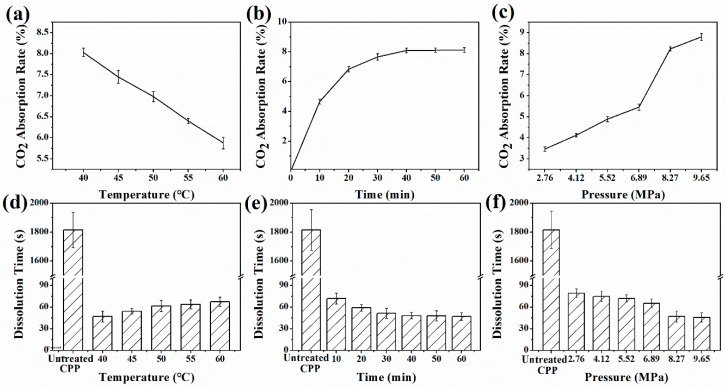
The SC-CO_2_ absorption rate of CPP solution (**a**) at different processing temperatures (codes 1–5), (**b**) for different processing time (codes 6–11), and (**c**) at different processing pressures (codes 12–17). The dissolution time of the prepared CPP (**d**) at different processing temperatures (codes 1–5), (**e**) for different processing time (codes 6–11), and (**f**) at different processing pressures (codes 12–17).

**Figure 5 foods-10-01555-f005:**
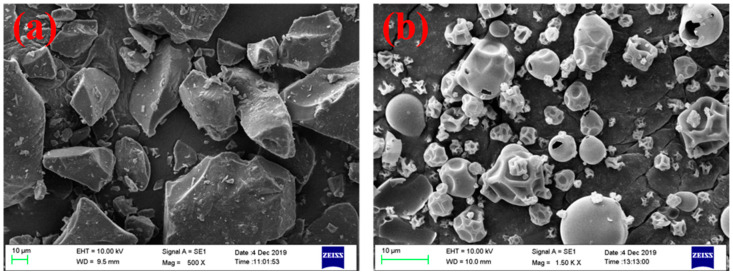
SEM images of (**a**) untreated CPP powders and (**b**) SAA-CPP powders.

**Figure 6 foods-10-01555-f006:**
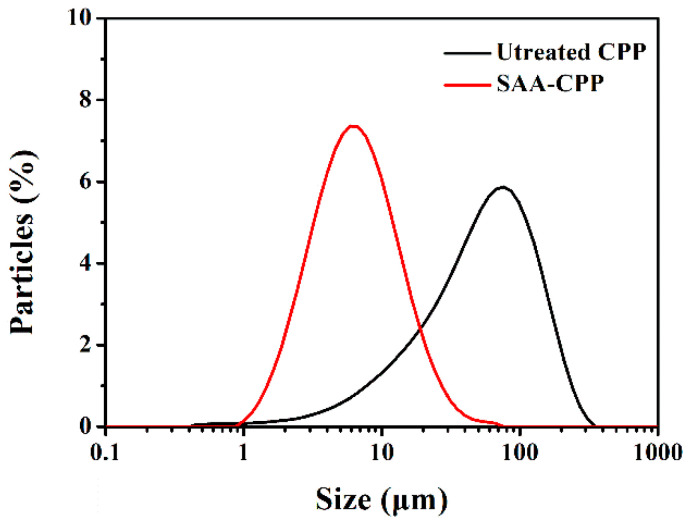
Particle size distribution of untreated CPP powders and SAA-CPP powders.

**Figure 7 foods-10-01555-f007:**
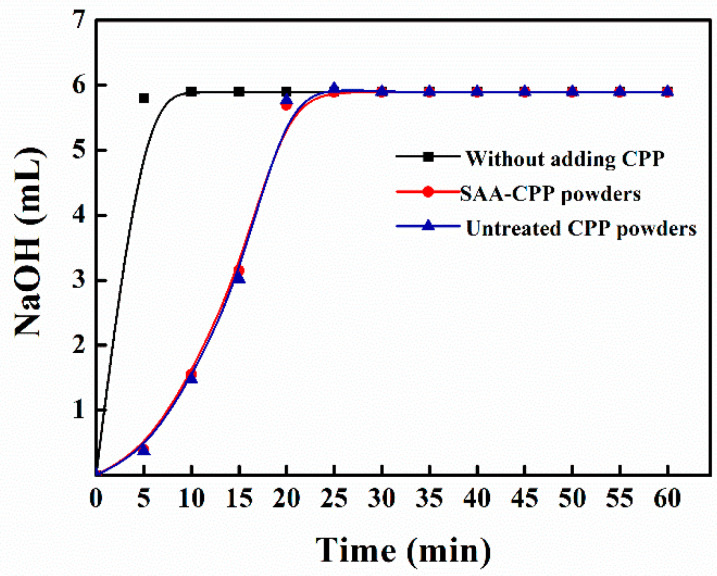
The calcium-binding capacity of the untreated CPP powders and SAA-CPP powders.

**Figure 8 foods-10-01555-f008:**
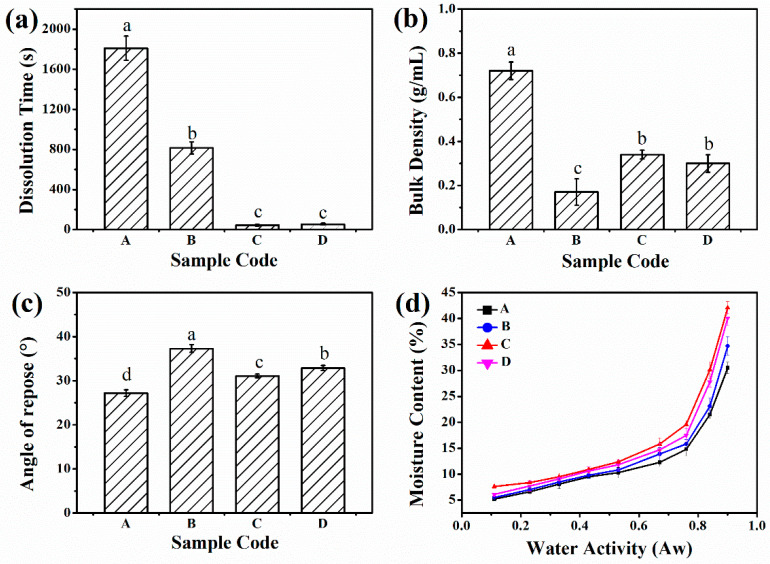
(**a**) Dissolution time (**b**) bulk density, (**c**) angle repose, (**d**) moisture contents of four different kinds of CPP powders (A: untreated CPP, B: spray-dried CPP, C: vacuum freeze-dried CPP, D: SAA-CPP); a–d means in the same figure with different superscripts differ significantly (*p* < 0.05) according to Duncan’s test.

**Table 1 foods-10-01555-t001:** Sample codes and processing conditions. “-” means without processing.

Sample Codes	Temperature (°C)	Time (min)	Pressure (MPa)
Untreated CPP	-	-	-
Effect of temperature, Tm			
1	40	40	8.27
2	45	40	8.27
3	50	40	8.27
4	55	40	8.27
5	60	40	8.27
Effect of time, Tt			
6	40	10	8.27
7	40	20	8.27
8	40	30	8.27
9	40	40	8.27
10	40	50	8.27
11	40	60	8.27
Effect of pressure, P			
12	40	40	2.76
13	40	40	4.12
14	40	40	5.52
15	40	40	6.89
16	40	40	8.27
17	40	40	9.65

**Table 2 foods-10-01555-t002:** The specific surface area, total volume in pores, and total area in pores of untreated CPP powders and SAA-CPP powders.

Samples	Specific Surface Area (m^2^/g)	Total Volume in Pores (cm^3^/g)	Total Area in Pores (m^2^/g)
Untreated CPP	0.263 ± 0.006 ^b^	0.00028 ± 0.00002 ^b^	0.098 ± 0.003 ^b^
SAA-CPP	2.40 ± 0.12 ^a^	0.00157 ± 0.00008 ^a^	0.963 ± 0.012 ^a^

All data are expressed as mean ± standard deviation of three measurements. ^a, b^ means values in the same column with different superscripts differ significantly (*p* < 0.05).

**Table 3 foods-10-01555-t003:** The physicochemical properties of the untreated CPP powder.

Samples	Hydroxyl Radical Scavenging Ability (U/mL)	Total Antioxidant Capacity (U/mL)	Foam Ability (%)	Foam Stability (%)	Conductivity (μS/cm)
Untreated CPP	97.5 ± 0.6 ^a^	2.22 ± 0.08 ^a^	60.2 ± 0.3 ^a^	53.3 ± 0.8 ^a^	1043 ± 2 ^a^
SAA-CPP	97.3 ± 0.8 ^a^	2.15 ± 0.09 ^a^	56.1 ± 0.3 ^b^	1.32 ± 0.14 ^b^	1039 ± 1 ^b^

All data are expressed as mean ± standard deviation of three measurements. ^a, b^ means in the same column with different superscripts differ significantly (*p* < 0.05).

## Data Availability

Not applicable.

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
