# Peer review of "Preparation and Characterization of Instant Casein Phosphopeptide by Supercritical Fluid Assisted Atomization"

_foods, 2021, doi:10.3390/foods10071555_

Round 1

Reviewer 1 Report

The manuscript is interesting and very very well written.  There are few minor suggestions/ corrections that could improve the manuscript.

line 86 please provide the purity of chemicals

line 176 provide %RH for these saturated solutions

the term “under” in conjunction with the pressures is confusing, please rephrase

line 272, 308 Term “blank” confusing, the term "untreated" much better

line 252 change "mutational site" to "inflection point"

line 238 the solubility of gases in liquids are inversely proportional to temperature

239 nothing magical about critical point of CO2

line 291 replace "absorption" with "dissolution"

line 329 omit “be used to”

Table 3 needs statistical letters denoting significant differences

Figure 8 stat letters denoting significant differences

Please provide discussion of overall comparison to other “standard" methods, e.g., cost & performance of product

Reviewer 2 Report

The manuscript is written well and it need to do a proper statistical analysis.
I would suggest the author do a one way ANOVA and Post Hoc test. 

Reviewer 3 Report

The authors present an interesting work on the preparation and characterization of instant casein phospho-2 peptide by SAA, however some improvements should be made before publication:

  • The main weakness of the manuscript is that the results section is fairly poor. No discussion of the results is made regarding similar findings of other authors. Only a few references are mentioned to have similar results (references 26, 30 and 31) but, even in these cases, no discussion is included. The authors should mention previous works regarding SAA, even if other substances are treated, and compare results.
  • The procedure for determination of SC-CO2 supersaturation in section 2.5 is not clear. further explanation or a reference with a more detailed description are needed.
  • What is the difference between blank and untreated CPP powders in Figure 7? In lines 119 and 120, the authors write: The commercial CPP powders without any treatment were used as the blank.
  • In table 3, it should be considered that standard deviations should have only one significant figure. Only when the leading digit is 1, the standard deviation can have two significant figures.
  • In Table 2, there are not standard deviations. Has only one measurement been made? No replicates?
  • Revise the numbering of Figures 8 in the text.

Round 2

Reviewer 3 Report

The authors have answered most of the questions I plotted, however, still some minor revision should be made:

  • In Figure 2. In MPa, p should be a-capital letter
  • When deciding the number of significant decimal positions, I suggest to use the recommendations made by Prof. Pasternack, from John Hopkins University. https://www2.chem21labs.com/labfiles/jhu_significant_figures.pdf Numbers such as 97.51±0.6 in Table 3 or 0.26±0.006 are not correct.
  • There is still some confusion with the blank. Review the sentence in lines 118-119 (The commercial CPP powders without any treatment were used as the blank) and Table 1.

Author Response

Point 1: In Figure 2. In MPa, p should be a-capital letter

Response 1: We have revised “p” to a capital letter “P” in Figure 2.

Point 2: When deciding the number of significant decimal positions, I suggest to use the recommendations made by Prof. Pasternack, from John Hopkins University. https://www2.chem21labs.com/labfiles/jhu_significant_figures.pdf Numbers such as 97.51±0.6 in Table 3 or 0.26±0.006 are not correct.

Response 2: Thank you for your suggestion. We have revised the significant figures in Table 2 and Table 3.

Point 3: There is still some confusion with the blank. Review the sentence in lines 118-119 (The commercial CPP powders without any treatment were used as the blank) and Table 1.

Response 3: Thank you for your suggestion. We have revised the sentence in lines 118-119 (The commercial CPP without any treatment was tested as a comparison and coded as Untreated CPP) and Table 1.